# The Significance of Flavonoids in the Process of Biological Nitrogen Fixation

**DOI:** 10.3390/ijms21165926

**Published:** 2020-08-17

**Authors:** Wei Dong, Yuguang Song

**Affiliations:** School of Life Science, Qufu Normal University, Qufu 273165, China; dwei@qfnu.edu.cn

**Keywords:** flavonoids, biological nitrogen fixation, endosymbiosis, nodulation, actinorhiza, arbuscular mycorrhiza

## Abstract

Nitrogen is essential for the growth of plants. The ability of some plant species to obtain all or part of their requirement for nitrogen by interacting with microbial symbionts has conferred a major competitive advantage over those plants unable to do so. The function of certain flavonoids (a group of secondary metabolites produced by the plant phenylpropanoid pathway) within the process of biological nitrogen fixation carried out by *Rhizobium* spp. has been thoroughly researched. However, their significance to biological nitrogen fixation carried out during the actinorhizal and arbuscular mycorrhiza–Rhizobium–legume interaction remains unclear. This review catalogs and contextualizes the role of flavonoids in the three major types of root endosymbiosis responsible for biological nitrogen fixation. The importance of gaining an understanding of the molecular basis of endosymbiosis signaling, as well as the potential of and challenges facing modifying flavonoids either quantitatively and/or qualitatively are discussed, along with proposed strategies for both optimizing the process of nodulation and widening the plant species base, which can support nodulation.

## 1. Introduction

The flavonoids form a large group of plant secondary metabolites synthesized by the phenylpropanoid pathway: over 9000 distinct compounds have been characterized to date [1]. The major recognized subgroups are the chalcones, flavones, flavonols, anthocyanins, proanthocyanidins and aurones [2] (Figure 1). A wide range of plant processes makes use of flavonoids, and these include protection from harmful radiation, sexual reproduction, defense against pests and pathogens and tissue pigmentation. Their synthesis involves a number of discrete enzymatic steps [3]. In the model angiosperm *Arabidopsis thaliana*, most of the relevant enzymes are encoded by a single-copy gene [4]. Flavonoids are synthesized in the cytosol [5] and are typically stored in the vacuole [6], but some are exuded into the rhizosphere [7]. The synthesis of a given flavonoid can be either organ- and/or tissue-dependent, and can be affected by the plant’s external environment, in particular by light intensity, ambient temperature and the availability of nitrogen [8,9]. The lateral root and nodule primordia of the legume subterranean clover (*Trifolium subterraneum*) is particularly rich in flavonoids [10], as is the root tip and lateral root primordia of *A. thaliana* [11].

The process of biological nitrogen fixation converts atmospheric nitrogen into ammonium, a form that is readily utilized by plants. The ability of some plant species to supply some, if not all of their requirement for nitrogen in this way gives them a substantial competitive advantage over those that lack this ability. According to some estimates, biological nitrogen fixation is responsible for the fixation in the agricultural system of up to 200 MT of nitrogen annually, representing a major saving in the cost (both financial and environmental) of crop production. The only micro-organisms able to carry out biological nitrogen fixation are those that produce nitrogenase, an enzyme that is required to catalyze the conversion of atmospheric nitrogen to ammonium. In a highly restricted group of plant species, the association between the host plant and the bacterial symbiont is a highly intimate one: the bacteria are housed within nodules, a specialized organ that forms in the root. Several nitrogen-fixing bacterial species are known to associate with non-nodulating plants; while these bacteria are generally free-living in the rhizosphere, in some cases they are able to colonize non-specialized intercellular spaces within the plant root [12]. The efficiency with which nitrogen is transferred to the plant by such bacteria is, however, relatively low, and the relationship between the two organisms is regarded as opportunistic rather than mutualistic. In genuine mutualistic symbioses, the host and symbiont appear to function essentially as a single organism [13].

**Figure 1 ijms-21-05926-f001:**
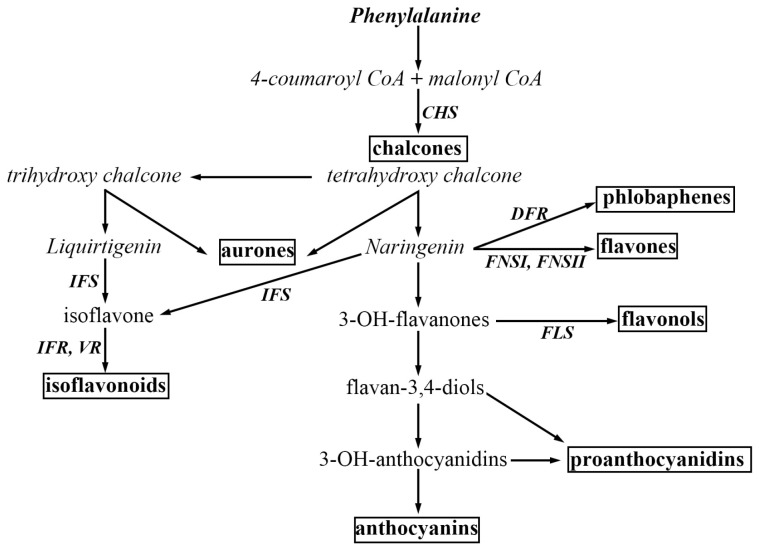
Major branches of the flavonoid biosynthesis pathway. Some of the critical enzymes are abbreviated as follows: CHS, chalcone synthase; DFR, dihydroflavonol 4-reductase; FSI/II, flavone synthase *I*/*II*; FLS, flavonol synthase; IFS, isoflavone synthase; IFR, isoflavone reductase; LCR, leucoanthocyanidin reductase; VR, vestitone reductase. Major classes of end-products are emphasized in boxes. This figure was adapted from Ref. [14].

Two types of intracellular endosymbiosis have been recognized, namely one which requires the formation of a root nodule, and one that relies on arbuscular mycorrhizae (AM) [15] (Figure 2A). The root nodule has evolved to facilitate both nitrogen fixation by the symbiont and the assimilation of ammonium by the host plant. Within the nodule, the symbiont receives its carbon and energy from the host and in return converts atmospheric nitrogen to ammonium, a process that requires an anaerobic environment. The bacteria capable of establishing this form of symbiosis belong to two distantly related clades, namely the proteobacterial *Rhizobium* spp. and the actinobacterial *Frankia* spp. Meanwhile the host species all belong to the so-called “nitrogen-fixing clade” [16], which consists of species within either the order Fabales (nodulated by the *Rhizobium* spp.) or the three orders Cucurbitales, Fagales and Rosales (nodulated by the *Frankia* spp.) [17]. Phylogenetic analyses suggest that all nodulating plant species belong to the Fabid (Eurosid 1) clade [16]. AM-based symbioses are, in contrast, very widespread in the plant kingdom, involving at least 80% of all angiosperm species; the microbial partners are not bacteria, but rather are fungi belonging to the phylum Glomeromycota [3]. They themselves do not fix nitrogen, but many studies have demonstrated that the presence of AM in the rhizosphere enhances the colonization of legume host roots with *Rhizobium* spp. Some features of root nodule endosymbiosis may have been recruited from the more ancient AM symbiosis [18] (Figure 2b), which has prompted the hypothesis that the two processes share aspects of the early signaling events [19]. Flavonoids are known to be required for the establishment of nodules in legumes, and are thus likely also to be important in both actinorhizal and AM symbioses [10]. This review aims to summarize our current understanding of the signaling and control of flavonoids in the biological nitrogen fixation process.

## 2. Flavonoids and Legume Symbiotic Nitrogen Fixation

The legume family (Fabaceae) is the third largest family of flowering plants, with members of more than 650 genera, 18,000 species spread around the globe [21]. Part of their evolutionary success is due to their symbiosis with the *Rhizobium* spp., which enables the plants to tolerate soils thatare deficient in nitrogen. In natural ecosystems, the quantity of nitrogen fixed by legumes is estimated to be 28–84 kg per hectare per year, while in a cropping environment, this can rise to several hundred kilograms per hectare [22,23]. Legume symbionts adopt one of three strategies to achieve biological nitrogen fixation, namely the Nod strategy, the T3SS (type III secretion system) strategy and the non-Nod/non-T3SS strategy (Figure 3). It was well verified that flavonoids released by the roots of legume species regulate the Nod strategy [10]. Besides that, flavonoids participate in several other different stages of the nodulation process, such as the chemoattraction of Rhizobium, the T3SS strategy, the development of nodule, the symbiont selection and so on.

### 2.1. Flavonoids Regulate the Expression of Nod Genes

A well-studied effect of root-exuded flavonoids is their regulation of *Rhizobium* spp. *nod* genes [24]. This function of flavonoids was discovered, dating back to 1986 when luteolin in *Medicago sativa* [25] and 7,4’ dihydroxyflavone in *T. repens* [26] were found to act as *nod* genes inducers. The concentration of flavonoid required for this induction is typically in the nanomolar to low micromolar range, and mixtures of different flavonoids can be more effective than a single compound. In most α- and β- proteobacteria rhizobial species, as has been reviewed elsewhere [27,28] (Figure 4), a number of *nod* gene products co-operate to synthesize the Nod factors required for nodule formation. These *nod* genes are regulated by NodD, a LysR family transcription factor. The binding of an appropriate flavonoid to NodD is thought to facilitate the access of RNA polymerase and thereby to enhance the transcription of the *nod* genes. The NodD-flavonoid complex binds to its target DNA sequences, known as a *nod* box. The perception of flavonoids by *Rhizobium* spp. is associated with a rise in the concentration of cellular calcium, which acts to induce the expression of *NodD* [29]. Secreted *nod* factors are recognized by plant LysM receptor-like kinases, triggering the characteristic curling of the root hair tip back on itself, thereby trapping the symbiont cells within a pocket, from which they are taken up into the root proper via an infection thread [30]. Once they reach the inner root, they are endocytosed into nodule cells and begin to fix nitrogen. Nod factors also induce cell division, as well as gene expression in the root cortex and pericycle, which initiates the development of the nodule [31,32].

### 2.2. Flavonoids Regulate the Expression of Genes Acting in the T3SS Strategy

Flavonoids also induce a number of genes responsible for the T3SS process (Figure 4) in some rhizobial species, such as *B. elkanii* USDA61 [23,33]. Most of these genes include a *tts* box *cis* element in their promoter region and their expression frequently depends on the presence of the TtsI transcription factor, which binds to the *tts* box. Expression of *ttsI* (as determined by promoter-lacZ reporter gene fusion constructs) is strongly induced by flavonoids in a NodD-dependent manner, because the promoter of *ttsI*, like that of the *nod* gene, contains the *nod* box [34]. The secretion of some *Rhizobium* spp. T3SS proteins—in particular certain Nops (nodulation outer proteins; also called T3 effectors)—is induced by flavonoids produced by the host [35,36]. These proteins act to suppress the host’s pathogen defense response and to promote several processes required to establish a symbiosis. Some Nops are thought to promote symbiosis more directly by interfering with the host’s nodulation signaling machinery [34]. Nops are passed from the *Rhizobium* spp. cells’ cytoplasm through the lumen of needle-like structures, which appear in electron micrographs as appendages referred to as T3 pili [37]. The capacity to form a viable symbiosis is compromised in mutant *Rhizobium* spp. strains deficient with respect to the synthesis of or secretion ability of T3 pili [38,39,40]. The expression of *Nops* genes, like that of most of the genes involved in T3SS, relies on the presence of NodD and particular host-derived flavonoids. For example, only when Rhizobium cultures are provided with flavonoids is it possible to immunologically detect the presence of both NopX and NolT [41] and for long T3 pili to form [42].

### 2.3. Flavonoids Act as Chemoattractants and Growth Stimulants for Rhizobium *spp.*

That flavonoids can act as a chemoattractant of *Rhizobium* spp. has been inferred from the observation that their abundance is high in the vicinity of the root tip [43,44] and particularly so near emerging root hairs, at which, *Rhizobium* spp. infection is initiated [45]. *Sinorhizobium meliloti* cells use flavonoids to promote their movement toward its host’s roots [46] and the concentration of flavonoid involved varies from 1 µM to as little as 0.1 nM, a much lower level than is required to induce *nod* genes. Flavonoids can also regulate the growth of *Rhizobium* spp. For example, the growth of both *Bradyrhizobium japonicum* and *S. meliloti* cells was enhanced by the provision of daidzein, luteolin-7xO-glucoside or quercetin-3-O-galactoside produced by alfalfa [47,48]. The inclusion of either host plant exudate or the flavonoids naringenin and apigenin in the medium used to grow *Bradyrhizobium* sp. in vitro significantly enhances cell multiplication [7]. A similar effect is exerted by a number of simple phenolic acids (p-coumaric, caffeic, protocatechuic, p-hydroxybenzoic and phenyllactic acids) present in the rhizosphere as flavonoid breakdown products [49].

### 2.4. The Influence of Flavonoids over Nodule Development and Number

Auxin is synthesized locally in the shoot apex, the leaf primordia and the developing seed, and is subsequently transported away from the site of its synthesis by polar auxin transport [10]. Treating some legume species with a synthetic auxin transport inhibitor has long been known to induce the formation of pseudo-nodules; these structures contain a peripheral vasculature, which does not extend into its distal region, a central zone and a diffuse meristem [50]. Flavonoids regulate both the transport and breakdown of auxin during nodule development (Figure 5). Thus, flavonoids can potentially act within the root to control nodule development and differentiation [51]. It has been suggested that Nod factor perception induces certain flavonoids that inhibit auxin transport, thereby promoting the localized accumulation of auxin at the nodule initiation site, leading to the initiation of nodule primordia [52]. This notion has been experimentally validated by silencing the gene encoding chalcone synthase in the *Medicago truncatula* root [53]. Silencing various branches of the flavonoid pathway in *M. truncatula* shows kaempferol to be the flavonoid most likely able to inhibit auxin transport during nodulation [54]. Whether auxin transport is regulated by the nodulation process, leading to the determination of nodules in, for example, soybean, remains unclear, but it is likely that other flavonoids (possibly isoflavonoids) are also involved [55]. Silencing of the isoflavonoid synthesis pathway in soybean altered auxin-inducible gene expression and auxin transport in the roots, but this effect can be overcome either by inoculation with a genistein-hypersensitive *B. japonicum* strain or by providing purified *B. japonicum* Nod signals [55]. The nature of how flavonoids affect auxin transport is not known, but the evidence from experiments using *A. thaliana* suggests that flavonoids affect the vesicular cycling of PIN family auxin transporters, possibly through interactions with other regulatory proteins such as phosphatases and kinases [56,57]. Abolishing the activity of isoflavone reductase in the common bean reduces the number of nodules formed, while simultaneously down regulating the gene *GH3* [58]. In *M. truncatula* abolishing the activity of chalcone synthase however, has no effect on lateral root development [59]. Auxin accumulation can also be influenced by the rate of its peroxidase-induced breakdown, a process that can be modulated by flavonoids. The isoflavonoid formononetin, which accumulates in the nodule primordia of white clover, accelerates auxin breakdown, while 7,4′-dihydroxyflavone (and its glycosides), which accumulate in the vacuoles of the cortical cells that later form the nodule primordia, inhibit its breakdown [60]. The differential ability of flavonoids and the availability of a large range of such metabolites give plants a means to regulate nodule development.

Flavonoids may also participate in the systemic regulation of the nodule number. Split root experiments have demonstrated that the content of isoflavonoid formononetin (and its glycoside ononin) is reduced in both *Rhizobium* spp.-induced and AM-induced symbioses in a systemic manner, suggesting that a related autoregulation signal affects their synthesis [67]. The exogenous supply of ononin is able to only partially restore nodulation and mycorrhization, which was taken to imply that flavonoids actively control symbiosis [67]. Similarly, a comparison of grafts involving a supernodulating soybean shoot or a wild type soybean shoot with a wild type common bean root has shown that the accumulation of isoflavonoids in the root is higher in the former case [68]. The exogenous supply of either daidzein or coumestrol increases the nodule number and enhances bacterial growth in vitro [68]. It is possible that systemically accumulated flavonoids control auxin transport in autoregulated roots, since in *M. truncatula*, auxin transport has been shown to be modulated during autoregulation [69].

### 2.5. The Contribution of Flavonoids to Symbiont Selection

The *Rhizobium* spp. legume symbiosis is highly host-specific: that is, a given *Rhizobium* spp. strain is only able to form a successful symbiotic relationship with a limited set of host plant species and vice versa [70]. The combination of host flavonoids appears to be a major determinant as to which *Rhizobium* spp. are able to successfully establish a symbiotic relationship. The affinity between NodD and flavonoids in part determines the host range: it has been shown that the NodD protein produced by a broad host range *Rhizobium* spp. strain interacts with a greater number of flavonoids than does NodD produced by a narrow host range strain [71]. For a given flavonoid to make a significant contribution to determining whether or not the host and the symbiont will be compatible, it has to be a strong inducer of *nod* genes, be represented in the host root exudate and be required for *Rhizobium* spp. infection and its synthesis should respond positively to the presence of the symbiont’s Nod protein [72]. In the soybean/*Bradyrhizobium* spp. symbiosis, genistein has been shown to be a strong and selective *nod* gene-inducer, activating NodD from *B. japonicum* but not *B. elkanii* [73].

The combination of flavonoids present in the root exudate of legume species acts as a selective agent for compatible symbiotic organisms. For example, medicarpin, a flavonoid produced by both *Trifolium* and *Medicago* species, exerts an inhibitory effect on incompatible bacterial strains [74]. Methoxychalcone is the strongest *nod* gene inducer identified in the *Medicago* root exudate. All four *M. truncatula* genes encoding chalcone-O-methyltransferase, the key enzyme required to synthesize methoxychalcone, are induced in the root hairs of plants inoculated with a compatible *Rhizobium* spp. strain [75]. However, in soybean, none of the six genes encoding chalcone-O-methyltransferase are induced in the root hairs of plants being colonized by *Bradyrhizobium* sp. [76]. The implication is that the synthesis of methoxychalcone is not a general response to *Rhizobium* spp. infection.

A further determinant of the host range of a *Rhizobium* spp. strain is associated with its surface polysaccharides and secreted proteins/T3SS [24]. Surface exopolysaccharides required for establishing a successful symbiosis can be modified by flavonoids either during or after the synthesis of exopolysaccharides [77]. Exposure of *R. fredii* cells to 1 μM genistein alters their exopolysaccharides, both quantitatively and qualitatively [78]. Flavonoids also regulate most of the genes utilized in the T3SS process. Low concentrations of certain isoflavonoids can induce resistance in compatible symbiotic bacteria to the potentially bactericidal phytoalexins present in some root exudates.

### 2.6. Flavonoids Act during Nodulation as Phytoalexins

Phytoalexins are low molecular weight antimicrobial compounds that are produced by plants as a response to biotic and abiotic stresses. Flavonoids can act as suppressors of rhizosphere micro-organisms competing with *Rhizobium* spp. for colonization. Some key flavonoids have been identified in both soybean and *M. truncatula* as being required for the initiation and progression of infection, acting as phytoalexins to reinforce specificity [72]. Several studies have shown that during nodulation, not only is the abundance of *nod* gene-inducing flavonoids increased, but is that of flavonoids endowed with antibacterial and/or antifungal activity. While the production of phytoalexins during nodulation may at first seem to be counter-productive, it appears that these phytoalexins are not part of a generalized defense response, and many of them are not inducers of *nod* genes [76]. Indeed medicarpin, for example, even acts to repress *nod* gene transcription [76]. Methoxychalcone is a potent antagonist of Gram-positive bacteria [79], while genistein possesses both antifungal and antibacterial activity [34]. The flavonoid and NodD1dependent secretion of Nops is in a sense a double-edged sword: on the one hand promoting the establishment of symbiosis with one legume species and on the other impairing it with a different host species [80]. The direct or indirect recognition by the host of NopP in *Sinorhizobium fredii* HH103 [81] can activate a plant defense response inhibiting the infection by *Rhizobium* spp. and subsequent nodule formation. This sort of response is likely similar to the effector-triggered immunity documented in certain plant–pathogen interactions [82]. The apparently universal role of flavonoids as phytoalexinsin plants suggests that, along with their role in determining the *Rhizobium* spp. host range, their role in defense has likely been a key driver in the expansion and diversification of the legumes.

### 2.7. Flavonoids in Symbiosis Quorum-Sensing

Quorum sensing is a system of stimulus and response correlated to population density. Many species of bacteria use quorum sensing to coordinate gene expression according to the density of their local population. The processes of nodulation, symbiosome development, exopolysaccharide production and nitrogen fixation all depend on the ability of *Rhizobium* spp. cells to accumulate in and around a host plant’s roots and nodules. The symbiont’s global profile of gene expression, including that of genes encoding the key components of nitrogen fixation [83], is strongly dependent on quorum sensing. A number of higher plant species are able to synthesize mimics of bacterial quorum sensing compounds, the best characterized of which are the acyl homoserine lactones (AHLs). Some of these mimic compounds have proven to be flavonoids [84], of which the most prominent example is naringenin. This flavonoid acts not just as an inducer of *nod* gene expression, but also as a strong inhibitor of quorum sensing in *Pseudomonas aeruginosa* [85], as well as in both *Escherichia coli* and *Vibrio fischeri* [86]. Experiments conducted in *M. truncatula* have demonstrated that certain bacterial AHLs are able to stimulate the production of AHL mimics [87]. The implication is that there may well be a link between a plant’s perception of AHL and the activation of its flavonoid pathway, while at the same time there may be a feedback mechanism in the bacterial species. Critical threshold concentrations in the rhizosphere of flavonoid mimics have yet to be defined. The boost in the synthesis of AHLs generated by exposure to *nod* gene-inducing flavonoids reported in three different *Rhizobium* spp. suggests that a level of coordination exists between *nod* gene induction and quorum sensing. The formation of biofilms relies both on effective quorum sensing and on the presence of certain bacterial surface components [88]. In *Sinorhizobium fredii*, the presence of *nod* gene-inducing flavonoids and the NodD1 protein is required for the transition of a biofilm monolayer into a microcolony [89]. The possibility that flavonoids mimic quorum sensing compounds and, when present at a relevant concentration, can activate gene expression in rhizosphere bacteria suggests a means of manipulating the ability of bacteria to colonize their host plant.

## 3. Flavonoids in Actinorhizal Plant Nitrogen Fixation

Only a single family of host plants have succeeded in evolving a symbiotic relationship with *Rhizobium* spp., whereas the symbiosis between the actinobacterium *Frankia* spp. involves plants belonging to eight families: namely the Betulaceae, Myricaceae, Rosaceae, Datiscaceae, Elaeagnaceae, Coriariaceae, Casuarinaceae and Rhamnaceae [90]. Except for herbaceous species belonging to the genus *Datisca*, these so-called “actinorhizal” plants are all woody shrubs or trees, many of which are adapted to highly marginal environments [91], such as the sandy dunes in Africa where *Casuarinaceae* species, through their association with *Frankia* spp., have been recorded as able to fix an average of 15 kg nitrogen per ha per year. In some temperate environments, the capacity of actinorhizal plants to fix nitrogen can reach 300 kg nitrogen per ha per year [91]. Analysis of *Frankia* spp. genome sequences has failed to reveal any evidence for the presence of *nod* gene clusters, which implies that their mode of symbiosis differs markedly from that used by *Rhizobium* spp. On the other hand, it is becoming clear that flavonoids are central to the process of actinorhizal nitrogen fixation.

### 3.1. Flavonoids May Act as Signals for Establishing Actinorhizal Symbioses

While the molecular basis of the involvement of flavonoids in actinorhizal symbiosis remains poorly understood, the general assumption is that a compatible interaction between *Frankia* spp. and an actinorhizal plant relies on an exchange of signals between the two partners, with some indication that flavonoids participate in signaling, at least during the initial stage of the interaction. Flavonoids accumulate inside the actinorhizal nodule. Nodule formation by Frankia on the roots of red alder (*Alnus rubra*) is promoted by irrigation with flavonoid-containing seed washes prepared from the host species [92], and these results were reinforced by treatment with the quercetin and kaempferol contained in black alder (*Al. glutinosa*) root exudates [93]. The curling of root hairs, a key early event in the establishment of a symbiosis, can be promoted by exposing *Frankia* spp. cells to a filtrate prepared from the roots of *Al. glutinosa* [94]. Direct evidence for the participation of flavonoids during the early stages of actinorhizal nodulation has been provided by showing that abolishing the activity of chalcone synthase (the first enzyme of the flavonoid pathway) in *Casuarina glauca* significantly compromises nodulation [15]. Exudate from the fruit of *Myrica gale* influences the transcription of 22 *Frankia* spp. genes, while inoculation of *M. gale* with *Frankia* spp. alters the level of transcription of several genes acting in the flavonoid synthesis pathway [95]. An analysis of a *C. glauca* root and nodule expressed sequence tag (EST) database has revealed the identity of eight genes encoding enzymes involved in the flavonoid synthesis [96].

### 3.2. Flavonoids May Contribute to the Determination of Host Specificity

Flavonoids extracted from fruit of *M. gale* enhance both the growth and the efficiency of nitrogen fixation achieved by a compatible *Frankia* sp. strain, but has a negative effect on an incompatible strain [95], which implicates flavonoids in the process of the host’s selection of a symbiont. Flavonoid-containing extracts of the root of *C. cunninghamiana* have been found to alter certain surface components of a compatible *Frankia* sp. strain in relation to infectivity [97], and some experimental evidence supports the notion that flavonoids are involved in both the chemoattraction and proliferation of *Frankia* spp. cells in the rhizosphere [98].

### 3.3. Flavonoids May Be Involved in Nodule Development and Function

In the *C. glauca*/*Frankia* spp. symbiosis, flavan class flavonoids are accumulated specifically in the nodule lobes [99]. Although the same compounds can be found in both nodules and non-infected roots, the amount of each flavan is much higher in the former. The application of in situ hybridization technology has established that transcripts of the gene encoding chalcone synthase accumulate in flavan-containing cells present at the apex of the nodule lobe. The significance of this compartmentation is not understood, but its development clearly requires the exchange of signals between the host and the symbiont. Similarly, in *Elaeagnus umbellata*, the abundance of transcript generated from the gene encoding chalcone isomerase is particularly high in nodules, increasing during nodule development [100].

The observations that the *C. glauca* auxin influx carrier gene *Aux1* is upregulated during actinorhizal nodule formation [101] and that auxin is produced by *Frankia* spp. [102] suggest that auxin influences the actinorhizal infection process. The accumulation of auxin within the *C. glauca* actinorhizal nodule is the result of both the localized expression of auxin transporter-encoding genes and the synthesis of auxin in planta by the *Frankia* spp. symbiont [103]. Metabolomic and transcriptomic analyses of *Datisca glomerata* have demonstrated an abundance of flavonols, which are particularly powerful inhibitors of auxin transport [17]. Analysis of nascent nodules formed during the *C. glauca*/*Frankia* spp. interaction has revealed that genes encoding isoflavonoids are prominently transcribed, suggesting that these compounds too are important for the nodulation process [104] and isoflavonoids also have the capability to control auxin transport. The possibility is therefore, that as in the legumes, flavonoids also act as auxin transport inhibitors during actinorhizal symbiosis, thereby promoting the localized auxin accumulation required for nodule development.

## 4. The Participation of Flavonoids in the Tripartite (Legume/AM/Rhizobium spp.) Symbiosis

Once the hyphae of AM penetrate a host plant root, they form ecto- or endomycorrhizal invasion structures [105]. The host root exudate, which is able to stimulate the germination of the AM’s spores, the branching of its mycelium and its colonization of the host, contains a number of flavonoids [106,107], at concentrations varying in the range 0.5–20 μM. The isoflavonoid coumestrol has been identified as a particularly active stimulator of hyphal growth [28], while a *M. truncatula* mutant that hyperaccumulates coumestrol is particularly strongly colonized by its AM symbiont [108]. In most herbaceous legume species, a tripartite symbiosis can form between the plant, *Rhizobium* spp. and AM [109] The effectiveness of this symbiosis in terms of nitrogen fixation is higher than that of the bipartite legume/*Rhizobium* spp. interaction [110,111]. According to FAO estimate, it has been estimated that 175 Mt nitrogen is fixed annually worldwide in this way, contributing materially to a reduction in dependence on synthetic fertilizers and to the sustainability of agriculture and agroforestry. Metabolite profiling of the roots of *M. truncatula* colonized by AM has revealed that flavonoids accumulate at various stages of the colonization process [112]. Similarly, an elevated abundance of transcript generated from genes encoding phenylalanine ammonia lyase and chalcone synthase can be detected in the roots of *M. trunculata* colonized by the AM species *Glomus versiforme* [113]. In *T. repens*, the flavonoid composition of extracts of the shoot and root of plants grown in the presence of AM differs markedly from that of plants grown in its absence [114]. It is possible that the tripartite symbiosis is effective in promoting nodulation because flavonoids induced by the AM partner stimulate the synthesis of Nod factors. Soybean plants respond to the presence of AM by boosting their production of daidzein [115], a compound that acts as a *nod* gene inducer [116]. The flavonoids genistein and coumestrol along with daidzein have been identified as potential signaling compounds regulating the establishment of the soybean/AM/*Bradyrhizobium* sp. tripartite symbiosis [117]. The synthesis of coumestrol (a *nod* gene inhibitor in *S. meliloti*) by *M. truncatula* is induced by the presence of AM symbionts [113]. However, evidence showed that although the profile of flavonoids produced by soybean plants supported by both *Rhizobium* spp. and AM responds to both symbionts: the accumulation of flavonoids (including those which induce *nod* genes and hyphal branching) is inhibited by the symbionts, even though co-inoculated plants enjoy a higher degree of nodulation than those inoculated with *Rhizobium* spp. [118]. The enhanced degree of symbiosis exhibited by tripartite interactions may, therefore, be more the result of their superior ability to take up nutrients than to any stimulatory effect of flavonoids.

In summary, it seems likely that changes in the flavonoid profile are responsible for the regulation of the initial phase of an AM association, and that a tripartite association can enhance the efficacy of the nitrogen fixation process carried out by legumes. The recent recognition that strigolactones, which are present in root exudates, can act as host-recognition signals for AM [119] casts some doubt over the assumption that flavonoids act as signaling molecules [120]. In addition, the colonization of mutualistic fungus *Phomopsis liquidambari* increases auxin signaling in *Arachis hypogaea* as well as nodulation and nitrogen fixation of the host. It may also be of value to examine the possibility that flavone compounds act as inhibitors of auxin transport.

## 5. The Potential of Manipulating Flavonoids to Improve Biological Nitrogen Fixation

Supplementation of inoculants with flavonoids is already in commercial use as a means of promoting legume–Rhizobium symbiosis [121]. For example, the product SoyaSignal^TM^ includes added genistein and daidzein, which induce the expression of *B. japonicum*
*nod* genes. The possibility of manipulating the root–rhizosphere interaction (and in particular biological nitrogen fixation) by modifying the plant’s flavonoids composition and/or content has been suggested as a fruitful line of research. For the moment though, the relevant technology has yet to be developed, awaiting a better definition of the nature of the interactions involved between the host plants, the various flavonoids and the relevant soil micro-organisms. Due to the biological complexity of the rhizosphere, it is possible that altering the content of a particular flavonoid will influence the performance of symbiotic and/or pathogenic soil micro-organisms, and even that of non-target plants; such unforeseen effects will need to be borne in mind and explored. While certain flavonoids may indeed enhance nodulation and/or mycorrhization, they may also have an effect on bacterial quorum sensing, plant–plant interactions and soil biochemistry. Catechin, as an example, has been suggested to negatively impact the quorum sensing of desirable micro-organisms, to the extent that it could represent a potent allelopathic signaling compound, acting to suppress plant growth [86,122]. Similarly, naringenin, a compound recognized as an inducer of *nod* gene expression in several *Rhizobium* spp., may affect the regulation of quorum sensing of non-target bacteria [123,124]. The isoflavonoids present in soybean root exudate attract not only the symbiont *B. japonicum* but also the highly damaging pathogen *Phytophthora sojae* [125]. Furthermore, the presence in the rhizosphere of flavonoid metabolic breakdown products has the potential to affect the activity and availability of the flavonoid itself and even to have a harmful effect on beneficial micro-organisms [126]. Given that flavonoids can modify the structure of the rhizosphere microbial community, an important direction of future research will be to apply DNA-based approaches, such as the high throughput sequencing of 16s rRNA, to track changes in species representation in the rhizosphere in response to the manipulation of flavonoids. At the same time these data could improve the level of understanding of the functional role of the various flavonoids in the activity of both symbiotic and non-symbiotic rhizosphere species.

The restricted host range characteristic of the legume-Rhizobium symbiosis has given rise to a great diversity of flavonoids and Nod factors, only few of which have been studied in any detail. Given that the legumes represent the third largest family in the plant kingdom, the potential number of distinct Nod factor/flavonoid combinations will likely be very large; thus the benefits to be gained from rhizosphere engineering could well be substantial. The realization of this potential will, however, require a step increase in research efforts directed at the identification of host determinant factors and a detailed characterization of the infection process. In contrast to the legume–Rhizobium interaction, the symbiosis between actinorhizals and *Frankia* spp. remains rather poorly characterized. Nevertheless, the overall similarity between these two processes implies that gaining a more profound understanding of the significance of flavonoids in the process of nitrogen fixation in the legumes will also likely shed light on the mechanisms underlying actinorhizal nitrogen fixation.

## 6. Conclusions

From the results above described, it is evident that even if the role of flavonoids have been well characterized in legumes nodulation with Rhizobium, several question marks still remain for their role in the legume/Rhizobium/AM tripartite symbiosis and actinorhizal symbioses (Figure 6). The discovery that actinorhizal species and legumes share a common symbiotic signaling pathway [127] suggests that a major research challenge will be to identify which flavonoids are in common to both the *Rhizobium* spp. and the AM symbiosis, and which are unique. Gaining this understanding could guide strategies based on manipulating the flavonoid pathway, aiming to either improve the efficacy of the natural symbiotic systems or even to transfer the ability to fix biologically nitrogen from legumes into cereals [128].

## Figures and Tables

**Figure 2 ijms-21-05926-f002:**
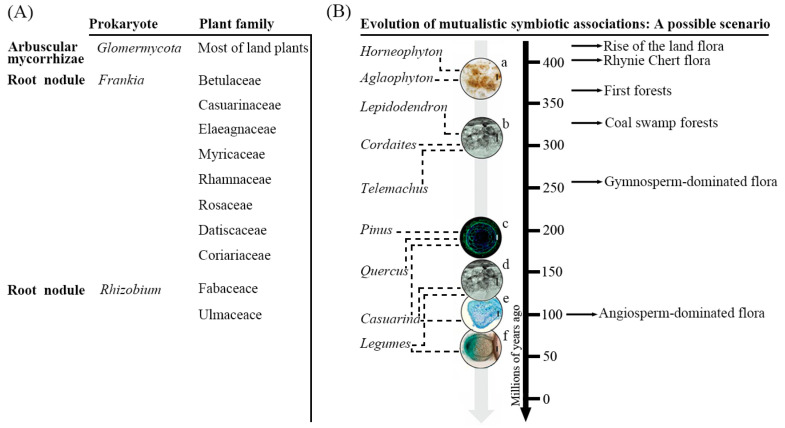
Plant families that participate in intracellular endosymbiosis (**A**) and the evolution of mutualistic symbiotic associations: a possible scenario (**B**) (modified from Martin et al., 2017 [20]); a, arbuscular mycorrhiza-like and/or mucoromycotina associations; b, arbuscular mycorrhiza; c, ectomycorrhiza; d, arbuscular mycorrhiza; e, Frankia N-fixing; f, Rhizobial N-fixing.

**Figure 3 ijms-21-05926-f003:**
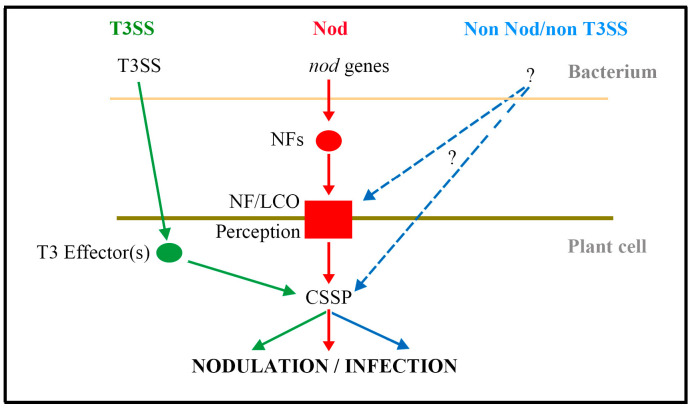
Nodulation strategies in legume symbionts (modified from Catherine and Joel, 2018 [23]). In the Nod strategy, strain-specific lipochitin oligosaccharides (LCO) called Nod factors (NFs) are produced under the control of *nod* genes. NFs are perceived by plant NF receptors that activate the common symbiotic signaling pathway (CSSP). In the T3SS strategy, T3SS effectors activate CSSP components by bypassing NF recognition. The mechanism of the third nodulation strategy is still unknown, but it involves neither *nod* nor T3SS functions and occurs via CSSP activation.

**Figure 4 ijms-21-05926-f004:**
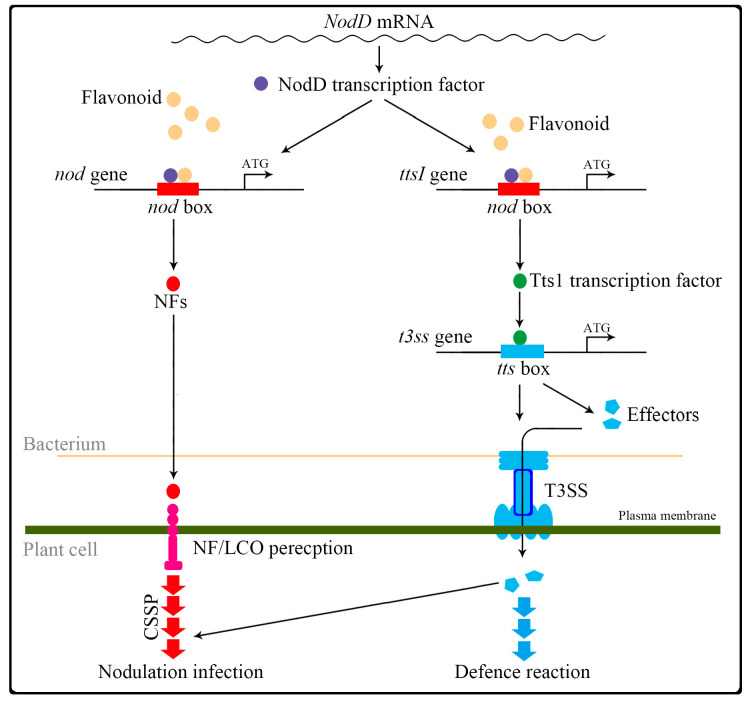
Contrasting cascades for the Nod and T3SS mechanisms.

**Figure 5 ijms-21-05926-f005:**
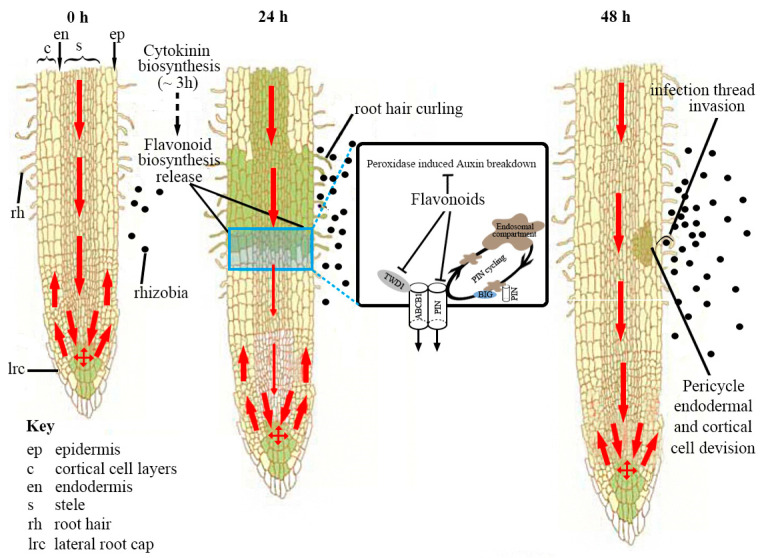
A schematic model of the regulation of auxin transport during nodulation in *Medicago truncatula*. Before rhizobia infection, auxin is transported in the acropetal direction towards the root tip. Auxin is also transported in the basipetal direction (from root tip to elongation zone) in the outer layer(s). Within 3 h after symbiosis induction (lipochitooligosaccharide treatment), cytokinin biosynthesis is upregulated in the *M. truncatula* roots [61]. Cytokinin perception at the inner cortex induces/releases certain flavonoids, which act as inhibitors of acropetal auxin transport at the inner cortical, endodermal and/or pericycle directly underlying the rhizobia infection site [62]. Flavonoids are auxin transport inhibitors that are thought to disrupt the complex between ABCB1 (ATP-Binding Cassette Subfamily B 1) and TWD1 (TWISTED DWARF1) [63,64], affecting transport, and by binding BIG, a protein required for PIN cycling [65]. The reduction in acropetal auxin transport increases the auxin concentration at the rhizobia infection site, the location of a future nodule primordium. An increase in basipetal auxin transport could also contribute to an increased auxin pool at the nodulation site [62]. Pericycle, endodermal and cortical cell divisions are activated within 48 h. The red arrow shows the polar auxin transport, and the arrow thickness is proportional to the auxin transport capacity. The green color shows the auxin gradient, and the darker color denotes a higher auxin content. This figure was adapted from Ref. [66].

**Figure 6 ijms-21-05926-f006:**
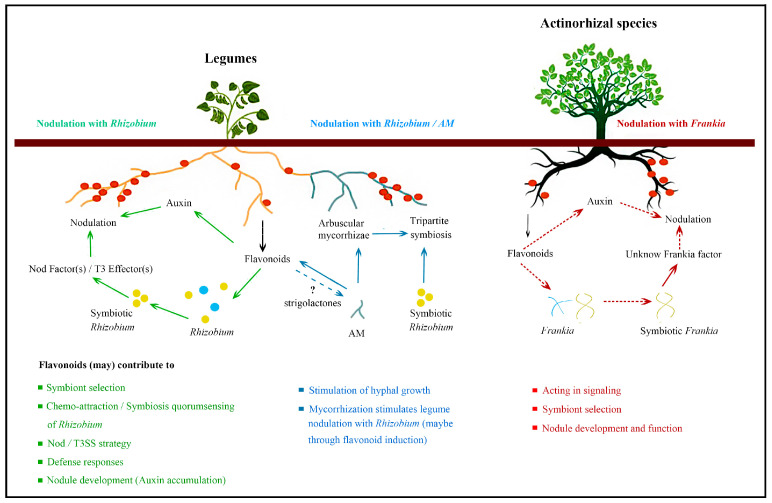
Schematic overview of flavonoid functions in the process of biological nitrogen fixation.

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
