# Peer review of "The Significance of Flavonoids in the Process of Biological Nitrogen Fixation"

_ijms, 2020, doi:10.3390/ijms21165926_

Round 1
Reviewer 1 Report
This is a very well-written and informative piece. However it is extremely dense and text-heavy and would benefit from additional figures (just two as submitted) and perhaps text boxes / glossaries. I believe these would make the review much easier to read, especially for the non-specialist.
Suggestions:
P2 - perhaps insert a figure showing the relative phylogenetic position of the 'nitrogen fixing clade' within the plant kingdom, showing how much more extensive and more ancient AM associations are than actinorhizal, and how much more widespread these are in turn than rhizobial associations
P4 - opportunity for a figure showing the contrasting cascades for the Nod and T3SS mechanisms
P5 - opportunity for a schematic figure showing how flavonoids can affect root and nodule development via modulation of auxin
P7- at the start of section 2.6 include a definition of phytoalexins (this is a review article so should be seen by a broader audience); similarly, at the start of section 2.7 explain what is meant by quorum sensing
P10-11 expand section 5. I would also like to see more forward-looking discussion, for example of further research questions that need answering, or real-world applications of the work (eg can you improve the N-fixation of leguminous crops by considering flavonoids as well as bacteria in seed-coatings and other inoculants)?
Corrections
P3 line 98 Figure 1 legend: correct to "modified from Catherine and Joel, 201821"
P3/P4 lines 111-114: this sentence is rather long and clumsy, breaking into smaller section would make it easier to read
P7 line 261 spelling mistake, should read 'phytoalexin'
P12 figure 2 - final point in green text should read 'auxin accumulation'
P12 figure 2 - final point in blue text contains several errors, perhaps should read 'Mycorrhization stimulates legume nodulation with Rhizobium (maybe through flavonoid induction)'
Reviewer 2 Report
The review paper report on the role of flavonoids in the three major types of root endosymbiosis responsible for biological nitrogen fixation and summarize the current understanding of the signaling and control of flavonoids in the biological nitrogen fixation process.
The manuscript is generally well written and clearly presented.
Major strengths.
- The review paper is of interest, actuality and use for this topic.
- Discussion explain what in the title is expected to evaluate.
- The topics are well explained and well documented with bibliographic references.
- The contribution to theory and practice would appear to provide useful information for researchers engaged worldwide in this research field.
I recommend publication after the Authors have considered some minor revisions (comments) here listed.
- Can you provide a briefly discussion/section of flavonoid biosynthesis pathway, especially the host plant specific steps? It could increase the interest of readers.
- “Conclusion” section: Can you provide more complete “Conclusion and Future Prospects” section? In this new way you can delineate and predict a deep knowledge, i.e., of specific host determinants, particularly the identification of infection flavonoids and the enzymes that produce them and their corresponding rhizobial NodD proteins, or the challenges offered by modern biotechonogies.
